# Pharmacological Approaches Using Diabetic Drugs Repurposed for Alzheimer’s Disease

**DOI:** 10.3390/biomedicines12010099

**Published:** 2024-01-03

**Authors:** Muna A. Adem, Boris Decourt, Marwan N. Sabbagh

**Affiliations:** 1Department of Neurology, Barrow Neurological Institute, St. Joseph’s Hospital and Medical Center, 350 W. Thomas Rd., Phoenix, AZ 85013, USA; 2Department of Pharmacology and Neuroscience, School of Medicine, Texas Tech University Health Sciences Center, Lubbock, TX 79430, USA; bdecourt@ttuhsc.edu

**Keywords:** Alzheimer’s disease, dementia, diabetes, type 2 diabetes, drugs, medications, clinical trials, insulin

## Abstract

Type 2 diabetes mellitus (T2DM) and Alzheimer’s disease (AD) are chronic, progressive disorders affecting the elderly, which fosters global healthcare concern with the growing aging population. Both T2DM and AD have been linked with increasing age, advanced glycosylation end products, obesity, and insulin resistance. Insulin resistance in the periphery is significant in the development of T2DM and it has been posited that insulin resistance in the brain plays a key role in AD pathogenesis, earning AD the name “type 3 diabetes”. These clinical and epidemiological links between AD and T2DM have become increasingly pronounced throughout the years, and serve as a means to investigate the effects of antidiabetic therapies in AD, such as metformin, intranasal insulin, incretins, DPP4 inhibitors, PPAR-γ agonists, SGLT2 inhibitors. The majority of these drugs have shown benefit in preclinical trials, and have shown some promising results in clinical trials, with the improvement of cognitive faculties in participants with mild cognitive impairment and AD. In this review, we have summarize the benefits, risks, and conflicting data that currently exist for diabetic drugs being repurposed for the treatment of AD.

## 1. Introduction

Alzheimer’s disease (AD) is a fatal, progressive neurodegenerative disorder characterized by the accumulation of amyloid-β (Aβ), neurofibrillary tangles (NFTs), and neuronal loss [1,2]. AD is the most common cause of dementia, affecting around 50 million people worldwide, with cases projected to increase to approximately 150 million by 2050 [3]. *APOE4* is the single strongest genetic risk factor for AD [4]. Approximately 25% of the population carries at least one ε4 allele [5], putting one fourth of the population at risk of developing AD. Diabetes mellitus is the most common chronic metabolic disorder. In the U.S., 38 million adults have diabetes [6], with type 2 diabetes mellitus (T2DM) making up 96% of the total cases [7]. This burden is only set to increase with the current sedentary lifestyles, rising obesity rates, and poor diet [8]. T2DM is characterized by hyperglycemia and insulin resistance [9,10]. T2DM is associated with a host of complications, including cardiovascular disease, stroke, chronic kidney disease, and vision loss [11]. Additionally, T2DM has been heavily associated with cognitive impairment [12,13] and dementia [14,15], namely, AD [16,17].

Following the initial Rotterdam study, in which there was a positive association between dementia and T2DM [18], additional epidemiological and clinical studies have been conducted in an attempt to establish a more concrete causal link between the two disorders [19,20]. Unfortunately, no one clear underlying mechanism linking the two has been uncovered yet. However, these links have introduced new avenues of investigations into AD drug research, one of which is the exploration of the role of insulin in the brain and the development of AD.

Insulin has neuromodulatory and neuroprotective effects on the brain [21]. This is in line with emerging research that has linked AD with common T2DM phenomena, such as insulin resistance, dysfunctional insulin signaling, neuroinflammation, advanced glycosylation end products (AGEs), and metabolic dysfunction [22], earning it the name “type 3 diabetes” [23]. In T2DM patients, insulin signaling is not processed correctly. In response to insulin resistance, pancreatic *β*-cells increase insulin production. Since the brain is a target organ for insulin, insulin signaling plays an important role in the organization and function of the brain. Impaired insulin signaling induces an overactivation of GSK-3 kinase, increases tau phosphorylation, alters tau modification, and neurofibrillary degeneration. This “type 3 diabetes” is a term proposed to describe the hypothesis that Alzheimer’s disease is caused by a type of insulin resistance and insulin-like growth factor dysfunction that occurs specifically in the brain. AGEs are implicated in AD pathogenesis through multiple mechanisms, such as accelerated Aβ deposition, increased APP expression, abnormal tau phosphorylation and oxidative stress [24,25]. These findings may act as an argument as to why insulin dysregulation, as is seen in T2DM, may play a role in the development of AD. This theory is taken a step further in animal models, where the induction of an insulin deficient state led to significant AD pathogenies, such as an increased Aβ burden [26,27,28] and the hyperphosphorylation of tau [29,30]. Interestingly, one such impaired enzyme system in diabetes-induced murine models were amyloid-beta degrading enzymes, namely, endothelin-converting enzyme 1 (ECE-1) and insulin-degrading enzyme (IDE) [31]. IDE has two substrates, insulin and amyloid beta, which play a key role in the pathogenesis of T2DM and AD [32]. Insulin resistance in the brain, mimicking peripheral insulin resistance in T2DM, can result in hyperinsulinemia in the brain, creating an environment in which insulin competes with Aβ as a substrate for insulin-degrading enzyme (IDE) [32].

These findings are further consolidated by studies in which insulin administration resulted in an improvement of cognitive function [33]. Additionally, the use of various other antidiabetic drugs has shown a similar beneficial effect on cognition [34] with a reduction in AD pathological burden [35,36]. To that effect, this review aims to examine and discuss the efficacy of the different animal and clinical studies conducted using a range of antidiabetic drugs such as metformin, insulin, incretins, dipeptidyl-peptidase 4 (DPP4) inhibitors, peroxisome proliferator-activated receptor γ (PPAR-γ) agonists, and sodium-glucose cotransporter 2 (SGLT2) in AD. These anti-diabetic medications are the most used in therapy according to the ADA and EASD recommendations. The results of this study suggest that antidiabetics may represent a promising alternative therapeutic approach for the treatment of AD.

## 2. Method

### Literature Search

All data were obtained from already existing literature on the electronic databases PubMed and Google scholar, using the following inclusion search criteria: “metformin Alzheimer trials”, “metformin amyloid beta”, “metformin tau”, “metformin clinical trials in Alzheimer’s”, “metformin neuroinflammation”, “Intranasal insulin Alzheimer trials”, “Insulin resistance in the brain”, “type 3 diabetes”, “insulin-degrading enzyme in Alzheimer’s”, “Incretins Alzheimer trials”, “GLP-1 analogues and Alzheimer’s” “GLP-1 analogues amyloid beta and tau”, “GLP-1 neuroinflammation”, “DPP4 inhibitors and neuroinflammation” “DPP4 inhibitors Alzheimer trials” “DPP4 inhibitors amyloid beta and tau”, “PPAR-γ agonists Alzheimer trials”, “PPAR-γ agonists beta amyloid”, “PPAR-γ agonists tau”, “PPAR-γ agonists neuroinflammation”, “SGLT2 inhibitor amyloid beta tau” “SGLT2 inhibitor Alzheimer trial”, “SGLT2 inhibitor AD pathology”, “T2DM and AD associations”, “Current therapeutic trials in AD”, “Current drug trials for AD”, “Antidiabetics in AD”. Studies from 1902 to 2023 were included. A total of 133 citations were identified. Studies were excluded if they did not meet the search criteria above. The drug classes are summarized in Table 1. 

## 3. Discussion

### 3.1. Metformin

Metformin, a biguanide derivative, is an insulin-sensitizing agent and the current first-line antidiabetic on the market [37]. The exact mechanism of action of metformin is unclear, but the main molecular mechanism it functions through is the inhibition of gluconeogenesis [38], favorably in the liver, due to the presence of OTC1 in hepatocytes [39]. The molecular mechanism of action of metformin seems to be tied to its inhibition of complex 1 in the respiratory chain [40,41], thereby inhibiting ATP production and increasing the AMP:ATP ratio [42]. This, in turn, activates AMP-activated protein kinase (AMPK), a cellular energy sensor [43,44], which functions to maintain energy homeostasis by activating catabolic pathways and inhibiting the anabolic pathways that consume ATP [45], such as gluconeogenesis.

Preclinical trials have demonstrated improved spatial memory in streptozotocin (STZ)-induced diabetic mice treated with 200 mg of metformin, likely through the promotion of the phagocytosis of amyloidogenic proteins, such as Aβ and NFT, as well as a reduction in neuronal death [46]. Another study is one investigating lysosomal autophagy pathways in AD mouse models, demonstrating that metformin acts as an activator of chaperone-mediated autophagy, a form of lysosomal autophagy that binds to and degrades amyloid precursor protein (APP), thereby increasing the clearance of APP and decreasing the accumulation of Aβ [47]. This study, [47], elucidates another molecular mechanism through which metformin may act to improve cognition and delay AD onset. In another study, metformin was shown to improve spatial memory and promote neurogenesis while decreasing neuronal loss, Aβ plaque load, and neuroinflammation in APP/PS1 mice [48]. Another study in which APP/PS1 mice were injected with tau aggregates and then treated with metformin showed that metformin stimulates the microglial-induced phagocytosis of amyloid deposits and tau proteins, thereby reducing amyloidogenesis in APP/PS1 mice [49].

Despite the robustly positive effect of metformin in preclinical trials, the results in clinical trials are more varied. For instance, a Singaporean study in which researchers collected data from 365 people aged ≥55 years from the population-based Singapore Longitudinal Aging Study with diabetes over a period of 4 years, found that the long-term use of metformin is associated with a decreased risk of cognitive decline (OR: 0.49 [CI 0.25–0.95]) [50]. A Finnish case–control study investigating the effect of the past use of metformin on clinically diagnosed AD has shown that not only does metformin exposure not result in AD, but also that long-term, high-dose metformin use is associated with a lower risk of incident AD in older people with T2DM [51]. These positive effects of metformin can also be applied to the general, nondiabetic population. For example, in a Mendelian randomization analysis of over half a million (527,138) individuals, genetically proxied metformin use demonstrated a 6.75 mmol/mol (1.09%) reduction of HbA1C was associated with lower odds of AD by 15% in the general population (OR: 0.85 [CI 0.78–0.93] *p* = 4.58 × 10^−4^) and by 4% in nondiabetics (OR: 0.96 [CI 0.95–0.98] *p* = 1.06 × 10^−4^) [52]. In a pilot study of individuals without AD, eighty men and women aged 55 to 90 years with amnestic mild cognitive impairment (AMCI) and without treated diabetes were randomized to metformin or placebo and observed for 12 months [53]. The study had two primary clinical outcomes: changes in the total recall of the selective reminding test (SRT) from baseline to 12 months and the score of the Alzheimer’s Disease Assessment Scale-cognitive subscale (ADAS-cog) [53]. After adjusting for baseline ADAS-cog, the metformin group showed improvements in the total recall of the selective reminding test (9.7 ± 8.5 vs. 5.3 ± 8.5; *p* = 0.02) [53]. Shockingly, a subgroup analysis of a meta-analysis of observational studies demonstrated that there is actually an increased risk of AD incidence in metformin users in Asians (OR 1.71 [CI 1.24–2.37] *p* = 0.001) [54]. These contrasting results could prompt further investigation into metformin as a possible agent in AD.

### 3.2. Insulin

Insulin is a 51-amino acid peptide hormone produced by pancreatic β-cells in response to elevated blood glucose. Insulin binds to the extracellular alpha subunit portion of the insulin receptor, a heterotetrameric tyrosine kinase receptor, inducing the dimerization of the intracellular beta subunits and receptor autophosphorylation [55]. This in turn will recruit and phosphorylate insulin receptor substrate (IRS) and activate the AKT pathway [55], which will result in the downstream activation of the master switches of cell metabolism and metabolic homeostasis such as glycogen synthase 3 (GSK 3) and mammalian target of rapamycin (mTOR) [56,57,58,59].

Insulin receptors are expressed all over the brain, with the highest density in the olfactory bulb, hypothalamus, hippocampus, cerebral cortex, and cerebellum [60,61]. Using a radioimmunoassay, insulin was initially detected in the brain by Havrankova et al. [62], who also determined in that same study that insulin is found in the brain at a much higher concentration than in the plasma. In a separate study using hyperinsulinemic (obese) mice and hypoinsulinemic (STZ-treated) rats, they determined that the brain insulin levels and brain insulin receptor concentration are independent of the plasma insulin levels and peripheral insulin receptor concentrations, indicating that the brain insulin systems are regulated independently of the peripheral insulin regulations systems [63]. At the one week and the one month mark, the STZ-treated rats showed no difference in the brain insulin despite total peripheral insulin depletion [63]. The obese mice were studied at the 8–10 week mark, showing markedly elevated plasma insulin levels with a reduction in peripheral insulin receptor expression, while their brain insulin remained at physiological levels and the insulin receptors remained similar to that of their thin counterparts [63].

Insulin resistance is defined as the body having an impaired response to insulin. This resistance developing in the brain could result in AD, and can serve as a possible link to T2DM. Many factors contribute to the development of brain insulin resistance, one of which is genetic polymorphism in the Fat Mass and Obesity-Associated Protein (*FTO*) gene [64]. A prospective cohort study showed us that carriers of *FTO* allele who were also carriers of apolipoprotein-E (*APOE*) ε4 allele have an increased risk of AD and dementia [65]. Fats are related to AD in other ways as well, with high-fat diets leading to the release of inflammatory markers at the hypothalamus, triggering the c-Jun N-terminal kinase (JNK) pathway to increase the activation of the leptin and insulin signaling inhibitor nuclear factor kappa-light-chain-enhancer of activated B cells [66,67]. The abnormal phosphorylation of IRS-1 has also been associated with brain insulin resistance.

Data from several human and animal studies have shown that the dysregulation of insulin function contributes to the development of neurodegenerative diseases [68]. In a study conducted using the homeostasis model assessment of insulin resistance (HOMA-IR) method and using verbal fluency as a measure of cognitive function, this impaired response to insulin was demonstrated to be linked to decreased verbal fluency, and thus increased cognitive decline and decreasing brain size and temporal gray matter [69]. Additionally, studies have indicated that brain insulin resistance and insulin-like growth factor 1 (IGF-1) resistance both play a large role in the development of AD [70,71], earning it the name type 3 diabetes. One study in particular demonstrated that insulin resistance in asymptomatic *APOEε4* carriers was found to be associated with higher levels of phosphorylated tau in the CSF [72], potentially indicating that insulin resistance plays a role in the phosphorylation of tau and may propagate the development of AD. This theory gains credence when studies using insulin in AD patients demonstrate a reduction in the hyperphosphorylation of tau and an increase in amyloid plaque clearance and synaptic plasticity [73,74]. To that effect, there has been a large interest in the use of insulin as a potential therapeutic agent for AD. 

One study using Tg2576 mice, which model AD-like neuropathology, to explore the link between insulin resistance and the development of AD [75] found that insulin resistance led to an increased deposition of amyloidogenic beta amyloid and decreased IDE function. IDE has two substrates that are important to us: insulin and Aβ. In one study, they demonstrated how IDE regulates Aβ levels in neuronal cells, with IDE knockout mice displaying hyperinsulinemia, glucose intolerance, and an increased accumulation of cerebral Aβ [32]. Another study demonstrated higher rates of Pittsburgh compound B (PiB) uptake in the frontal and temporal areas in patients with higher insulin resistance, correlating to increased amyloid in those regions [76].

In vivo investigations of insulin via peripheral modes of delivery is limited due to the risk of hypoglycemic events, so in its stead we use intranasal insulin for direct delivery and to bypass the periphery in its entirety. Intranasal insulin goes through the nasal passages and reaches the frontal cortex and the hippocampus within 15 min [77]. In 2008, Reger et al. demonstrated that administering 20 IU intranasal insulin twice a day for 21 days improves attention, story recall, and function in those with MCI or AD [78]. In another study, the cognitive dose–response curves of intranasal insulin were examined and researchers uncovered a difference in the response to insulin between the participants who were *APOE* ε4 carriers (ε4+) when compared to those without *APOE* ε4 (ε4−) [79]. The ε4− participants demonstrated an improvement in memory with insulin administration, whereas the ε4+ participants had a worsened cognitive course in comparison [79]. In 2012, Craft et al. ran a longer pilot clinical trial to investigate the effect of intranasal insulin on cognition, function, cerebral glucose metabolism, and CSF biomarkers in 104 adults with either AMCI (*n* = 64) or mild to moderate AD (*n* = 40) over the span of 4 months [80]. The participants were split into three groups receiving placebo (*n* = 30), 20 IU intranasal insulin (*n* = 36), or 40 IU intranasal insulin (*n* = 38) with the primary measures set as delayed story recall and Dementia Severity Rating Scale score [80]. The outcomes demonstrated an improvement in delayed memory in the 20 IU group (*p* < 0.05), as well as preserved caregiver-rated functional status at both insulin doses (*p* < 0.01) and general cognition assessed by the ADAS-cog score in the younger participants and functional abilities assessed by the ADCS-ADL scale for adults with AD (*p* < 0.05) [80]. Interestingly, the changes in memory and function in this study were associated with changes in the Aβ level as well as changes in the tau protein/Aβ42 ratio in the CSF [80]. Additionally, this study showed us that prolonged intranasal insulin use is not associated with any adverse events [80].

Most clinical trials investigating insulin use in MCI and AD use regular insulin analogues, so in 2015, Claxton et al. conducted a clinical trial using detemir, a long-lasting insulin analogue, to determine its effect on cognition and daily function in 60 adults with MCI or AD [33]. The participants were divided into placebo (*n* = 20), 20 IU of detemir (*n* = 21), and 40 IU of detemir (*n* = 19) and observed over a span of 3 weeks, after which there was improved verbal memory (*p* < 0.03) and visuospatial memory (*p* < 0.04) in the 40 IU group [33]. There was also improvement in cognition in the 40 IU group compared to the placebo group (*p* < 0.05); however, this result was affected by the *APOE* status, with improvement in cognition in the ε4+ participants (*p* < 0.02) and worsening in the ε4− participants (*p* < 0.02) [33], directly contrasting the results conducted in an earlier 2010 study [79].

In 2017, a longer pilot clinical trial comparing regular insulin to insulin detemir was conducted, where 36 adults diagnosed with MCI or AD were randomly assigned to placebo (*n* = 12), 40 IU of regular insulin (*n* = 12), or 40 IU of insulin detemir (*n* = 12) daily, over a 4-month period [81]. The regular insulin group displayed an improvement in memory, the primary outcome, at the two and four month marks compared to the placebo group (*p* < 0.03); meanwhile, the insulin detemir group had no significant changes from baseline compared to placebo group [81]. The regular insulin group also displayed the preservation of brain size on MRI as well as a decreased tau-P181/Aβ42 ratio [81].

In an attempt to further clarify the mechanism underlying the benefit of insulin in AD, Kellar et al. conducted a study in 2021 examining the effect of intranasal insulin on white matter health, cognition, and CSF biomarkers in adults with MCI or AD [82]. A total of 49 participants were randomized into a placebo group or an insulin group, receiving either placebo or 20 IU insulin twice daily for 12 months, and the researchers found that the insulin group displayed a decrease in white matter hyperintensity and global brain volume [82]. Comparatively, when intranasal insulin was used in a randomized clinical trial of 289 adults with MCI or AD over a span of 12 months, it showed no cognitive or functional benefits compared to the placebo group [83]. The results of this study is likely impacted by the fact that the insulin device was changed during the course of the trial [83]. Further studies should be conducted on insulin but, as it stands, it seems to show incredible therapeutic potential for AD.

### 3.3. Incretins

Glucagon-like peptide-1 (GLP-1) is an intestinal-derived incretin hormone. It was first discovered in 1902 when Bayliss and Starling fed ground up intestinal “extracts” to animals, after which they noticed a “reflexive” spike in pancreatic secretion followed by a drop in blood glucose [84]. These findings would be further elucidated upon in the century to come by Brown et al. [85]. These incretins are secreted in response to glucose, inducing glucose-dependent insulin secretion from pancreatic beta cells and the suppression of glucagon, resulting in lowered postprandial serum glucose [86,87]. Interestingly, GLP-1 receptors (GLP-1R) have also been found in the brain, specifically in the hippocampus, hypothalamus, cerebral cortex, and olfactory bulbs [88], thus increasing interest in the use of GLP-1 analogues as a potential therapeutic in AD. Furthermore, GLP-1 agonists have demonstrated neurotrophic and neuroprotective effects, likely through the promotion of long-term potentiation and synaptic growth [89]. They exhibited rescued cognitive function, decreased plaque burden, synaptic loss, and neuronal inflammation [90]. They also protect neuronal hippocampal cell death from Aβ1-42 [91], reduce APP and Aβ levels [92], and reverse AGE-induced tau hyperphosphorylation via the downregulation of GSK3β [25]. 

In APP/PS1 mice, liraglutide (LRGT), a GLP1-R agonist, prevented memory impairment, synapse loss, reduced β-amyloid plaque load and microglial-induced inflammation, and enhanced synaptic neuroplasticity [93]. Hyperhomocysteinemia is an independent risk factor for AD [94], and so injecting rats with homocysteine creates in them deficits and pathologies seen in AD [95]. LRGT use in homocysteine-treated rats resulted in the restoration of protein phosphatases-2A (PP2A) and demonstrated an inhibitory effect on β-secretases and γ-secretases, thereby reducing the production of Aβ and decreasing disease burden [96]. A study in which subcutaneous injections of LRGT were administered once daily for 8 weeks to mice prevented memory impairment, neuronal and synaptic changes, and resulted in the reduction in tau hyperphosphorylation via the protein kinase B and glycogen synthase kinase-3β (GSK3β) pathways [97]. One study in 2018 detailed how LRGT decreases tau pathology, reverses cognitive impairment in mice, and has a protective effect on insulin receptors and synapses in the brain via the activation of the protein kinase A (PKA) signaling pathways [34]. They did this by administering amyloid-β oligomers (AβOs) to non-human primates and tracking the loss of insulin receptors and synapses in the brain that followed on neuronal culture; whereas, LRGT-treated non-human primates displayed a preservation of the insulin receptors and synapses in comparison to the control group [34]. One study in particular demonstrated LRGT neuroprotective effects via the activation of phosphoinositide-3 kinase/mitogen-activated protein kinase (PI3K/MAPK) dependent pathways, resulting in the increased clearance of Aβ by increasing Aβ transporters in the CSF [98]. Another study focusing on the anti-AD effects of LRGT attempted to further elucidate the pathways through which LRGT may be ameliorating AD-related neurodegeneration [99]. In this study, the researchers used blood and brain cortical lysates obtained from triple transgenic-AD (3xTG-AD) female mice treated with peripheral LRGT for 28 days and evaluated for parameters affected by AD such as Aβ and p-tau, motor and cognitive function, glucose metabolism, inflammation, and oxidative/nitrosative stress [99]. LRGT was found to activate PKA pathways, oxidative/nitrosative stress, and inflammation in these mice, while reducing their cortical Aβ1–42 levels [99]. In 2021, another study set out to demonstrate the LRGT effects on mice with coexisting T2DM and AD (APP/PS1xdb/db mice) over a period of 20 weeks [100]. The results showed that LRGT caused the marked reduction in brain atrophy in the diabetic (db/db) and the APP/PS1xdb/db mice, as well as reduced Aβ aggregates levels (*p* = 0.046) and tau hyperphosphorylation (*p* = 0.009) in the APP/PS1xdb/db mice [100]. There was also rescued cognition in APP/PS1xdb/db mice, as was demonstrated by the new object demonstration test (*p* < 0.001) and the Morris water maze (*p* < 0.001) [100]. A number of other animal studies have been conducted, clearly delineating the positive effect of LRGT on AD pathology [101,102,103]. ELAD, Evaluation of Liraglutide in the treatment of Alzheimer’s Disease, was a phase IIb double blinded, randomized, placebo-controlled trial conducted in multiple centers in the UK, where 204 adults with mild to moderate AD received subcutaneous injections of either LRGT or placebo once daily for 12 months [104]. The results demonstrated no difference between the treatment and control in terms of the cerebral glucose metabolic rate, the primary endpoint [104]; however, there was improved cognitive function in the LRGT-treated participants, measured by ADAS-EXEC (ADAS-Cog with Executive domains of the Neuropsychological Test Battery).

Semaglutide exhibited pro-autophagy via the increased expression of LC3II, Atg7, Beclin-1, and P62, as well as an anti-apoptotic effect via the inhibition of the Bax system that was induced by Aβ25-35 in an AD model (SH-SY5Y cells with Aβ25-35) [105]. Currently, two large phase III clinical trials, evoke and evoke+ are underway [106]. Each study has 1840 amyloid-positive participants with MCI or mild AD dementia who will be randomized to receive either daily oral semaglutide (14 mg, escalated via 3 and 7 mg over 8 weeks) or daily oral placebo over a period of 156 weeks [106], with both trials set to be completed in September 2025. The difference between the studies is the inclusion of participants with vascular co-pathologies in evoke plus [106].

In one study, researchers injected exendin-4 (EX-4), a GLP-1R analogue, into transgenic *C. elegans* and observed the amelioration of Aβ1-42 toxicity via an EX-4 antioxidant effect through DAF-16 as well as its reduction in Aβ1-42 expression and accumulation [107]. Another study demonstrated the neuroprotective and neurotrophic effects of EXE-4 in the brain of mice that had undergone mild traumatic brain injury [108]. In 2023, Zago et al. set out to investigate the effects of EX-4 on the memory and hippocampal neurons of rats with sporadic dementia of the Alzheimer’s type (SDAT), in which the STZ-treated male Wistar rats were treated with EX-4 over a span of 21 days, during which memory and learning were assessed using a Y-maze (YM), object recognition tasks (ORTs), and object displacement tasks (ODTs) [109]. The results showed that the agonists of GLP-1R are anti-apoptotic, encourage the proliferation of hippocampal neurons, and preserve memory [109]. Secondary outcomes from a small (*n* = 18) pilot clinical trial using Exenatide, the synthetic form of EX-4, revealed no benefit of exenatide; however, no firm conclusions can be drawn from this study due to its early termination [110]. The REWIND trial was a randomized, double-blind placebo-controlled trial conducted in 24 countries which examined the effect of once weekly subcutaneous injection of either Dulaglutide (DGT) or placebo in participants aged 50 or more and diagnosed with T2DM on the cardiovascular risks of T2DM, such as non-fatal MI, non-fatal stroke, or death from cardiovascular causes [111]. An analysis of the cognitive impairment experienced by the participants was conducted using the Montreal Cognitive Assessment (MoCA) and Digital Symbol Substitution Test (DSST) at baseline and then at follow-up to assess cognitive impairment [112]. After adjustment, the hazard of substantial cognitive impairment was reduced by 14% in the DGT-treated arm in comparison to the placebo arm (HR 0.86 95% CI: 0.79–0.95 *p* = 0.0018), indicating that DGT may be a potential drug used to curb MCI in T2DM [112].

### 3.4. DPP4 Inhibitors

Dipeptidyl-peptidase 4 (DPP4) inhibitors increase GLP-1 levels, through which they function to decrease glucose levels to treat T2DM [113]. According to one study, linagliptin, a DPP4 inhibitor, can ameliorate neurodegenerative effects via insulin signaling [114]. Linagliptin treatment for 8 weeks was also found to improve brain incretin levels, while also reducing Aβ load, tau hyperphosphorylation, and neuroinflammation in 3xTg-AD mice [115]. Another molecular pathway to consider would be GSK3β, which has been implicated in AD pathogenesis [116], primarily through the hyperphosphorylation of tau, reduction in acetylcholine synthesis, and elevation of Aβ production. With that in mind, the multiple beneficial effects of DPP4 inhibitors on GSK3β should not be discredited. An example of this is one study in which linagliptin was found to restore the impaired downstream insulin signaling induced by β-amyloid in neurons, which in turn prevented the activation of GSK3β and tau hyperphosphorylation [117]. These findings highlight the significant role DPP4 inhibitors may play in the neurotoxicity of AD. DPP4 inhibitors have been shown to reduce pancreatic beta cell apoptosis via the suppression of the endoplasmic reticulum stress-mediated apoptosis pathway in diabetic mice [118]. One component of the ER stress-mediated apoptosis pathway, C/EBP homologous protein (CHOP), has been found to increase Aβ levels, induce reactive oxygen species accumulation, and promote neuroinflammation [119]. Henceforth, DPP4 inhibitors’ downregulatory effects on these stress proteins could serve as an alternate insight of its beneficial effect in AD pathogenesis. 

Daily, long-term (12 weeks) treatment with sitagliptin in an AD mouse model has resulted in delayed amyloid deposition, reduced ROS and neuroinflammation, and reduced beta amyloid burden [120]. A later study demonstrated a similar effect of other DPP4 inhibitors, saxagliptin and vildagliptin, in which the DPP4i-treatment of an STZ-induced rat model of AD led to a decrease in Aβ, t-tau, p-tau levels, and neuroinflammation, with an improvement in hippocampal memory retention [121,122]. One study of the long-term (2 years) use of DPP4 inhibitors resulted in preserved cognition in diabetics with MCI [123]. This study can also be used as evidence of the safety profile of chronic DPP4 inhibitor use as none of the participants reported any adverse effects after long-term treatment [123]. In another clinical study, where 253 elderly participants with T2DM were assigned to a sitagliptin or non-sitagliptin group, reduced insulin dosage and increased Mini Mental State Exam (MMSE) scores were observed in the sitagliptin group in comparison to the non-sitagliptin group [124], indicating improved cognition with sitagliptin treatment in elderly individuals with or without AD. The above data suggest promising therapeutic potential of DPP4 inhibitors in the treatment of AD through the targeting of core pathological features such as Aβ production, tau hyperphosphorylation, synaptic loss, and neuroinflammation.

### 3.5. PPAR-γ Agonists

The peroxisome proliferator-activated receptor γ (PPAR-γ) is a ligand-activated nuclear receptor that coordinates lipid and glucose metabolism and cellular homeostasis [125]. The two major PPAR-γ agonists are pioglitazone (PGZ) and Rosiglitazone (RSG). There has been found the increased expression of PPAR-γ in the temporal cortex of AD patients in comparison to control group [126], marking them as a potential therapeutic target in AD.

PGZ decreased extracellular Aβ1–42 levels in hamster ovary cells transfected with mouse APP 695 [127]. A study on mouse neuronal structures and human neural cell lines (SH-SY5Y) demonstrated that activated PPAR-γ protects neurons from APP misfolding, tau hyperphosphorylation, and synaptic loss [128]. Low-dose PGZ over seven weeks has been shown to improve learning and memory in senescence-accelerated mouse prone-8 (SAMP8) mice via the upregulation of lipoprotein receptor-related protein 1 (LRP1), which upregulates the clearance of Aβ [129]. The hyperactivation of cyclin-dependent kinase-5 (Cdk5), a serine/threonine kinase, plays a role in neurodegenerative processes, including those involved in the pathogenesis of AD [130,131]. PGZ functions via the inhibition of Cdk5, which interferes with the expression of PPAR-γ targets and therefore results in the increased degradation and overall reduction in Aβ levels [132]. Another study that supports this finding is a meta-analytical study of PPAR-γ agonists in AD, which demonstrated that PGZ improved the synaptic defects in AD transgenic mice via the inhibition of Cdk5 [133]. PGZ was also found to normalize p35 protein and CRMP2 levels in the cerebellum, with the improvement of coordination and long-term depression in APP/PS1 mice [134], suggesting PGZ as a prophylactic to be used at the pre-Aβ accumulation stage in AD model mice. Another molecular mechanism through which PGZ works is the regulation of AKT/GSK3β activation, resulting in improved peripheral and central insulin sensitivity, increased Aβ42 degradation, and decreased Aβ accumulation in diet-induced insulin resistance rats [135]. Conversely, a study in which P301S mice used as tauopathy models were treated with either PGZ or placebo over 6 months showed us that PGZ only altered the time course of microglial activation but did not significantly affect microglial activation in response to tau [136]. A pilot study conducted in AD and MCI participants with T2DM improved both the cognition and metabolic profile of the participants in the PGZ group [137]. Another clinical trial was conducted over a period of six months; a randomized, open-controlled trial in AD-T2DM participants, which resulted in the improvement of cognition and cerebral blood flow in the PGZ group compared to the control group [138].

Chronic RSG treatment in an AD mouse model prevented and reversed memory impairment, seemingly through the prevention of the downregulation of glucocorticoid receptors in the hippocampus [139]. Another study demonstrated RSG effectiveness in rescuing memory impairment through decreasing Aβ burden, decreasing neuropril containing phosphorylated tau, decreasing inflammatory markers, the activation of microglial-induced phagocytosis and increasing the clearance of Aβ in transgenic mice [140]. RSG has been shown to improve cognitive impairment in AD patients and improves deficits in the Tg2576 mouse for AD amyloidosis [141]. A small (*n* = 30) placebo-controlled, double-blinded parallel-group pilot study conducted in MCI and AD, where the participants were assigned to daily RSG (*n* = 20) or placebo (*n* = 10) for six months concluded with an improved attention and delay recall in the RSG group compared to the placebo group [142]. A larger study was conducted to assess varying dosages of RSG (2,4, or 8 mg) in comparison to placebo, with stratification by the *APOE4* status in the participants with mild to moderate AD over a span of 24 weeks [143]. A significant improvement in ADAS-Cog was observed in the *APOE4*-negative participants receiving 8 mg RSG, whereas no improvement and, at lower doses of RSG, the worsening of cognitive function was observed in the *APOE4*-positive treatment group [143]. Another study in which the effects of metformin, RSG or a combination of the both on cognitive impairment was evaluated, with RSG demonstrating a superior cognitive sparing function in older individuals with T2DM and MCI in comparison to the metformin group [144]. Contrarily, an earlier phase III, randomized, double-blinded placebo-controlled study of extended-release RSG (RSG XR) with prospective stratification by *APOE4* status was conducted to confirm the efficacy and safety of RSG XR in mild to moderate AD showed no benefit with the use of RSG XR [145]. Despite overall encouraging results, there remain glaring limitations to Thiazolidinedione’s use in AD. PGZ has a limited penetration into the brain [146] as well as the many side effects that were reported with RSG use [147], leading to its restriction by the FDA and suspension by the EMA in 2010. These risks in combination with limited beneficial clinical data restrict PPAR-γ agonists’ use in AD for the time being.

### 3.6. SGLT2 Inhibitors

Sodium-glucose cotransporter 2 (SGLT2) inhibitors block renal glucose, promoting glucosuria and thereby lowering blood glucose levels [148]. These drugs have been approved for use in a multitude of diseases, such as TD2M [149], decreasing the risk of major cardiovascular events in patients with T2DM and established cardiovascular disease [150], decreasing the risk of eGFR decline and the hospitalization of patients with chronic kidney disease [151], decreasing the morbidity and mortality in heart failure with a reduced ejection fraction (NYHA class II-IV) [152], and the improvement of heart failure with preserved ejection fraction [153]. These transporters have been found in the mammalian brain [154,155], and the use of SGLT2 inhibitors have demonstrated a neuroprotective effect [156,157] via improved mitochondrial function, preserved synaptic plasticity, as well as decreased insulin resistance, inflammation, and apoptosis. Empagliflozin has also been shown to increase the levels of cerebral brain-derived neurotrophic factor (BDNF) in db/db mice, which ensures neuronal growth, survival, and plasticity [157]. BDNF is also an important factor in learning and memory [158]. These findings make SGLT2 inhibitors an attractive therapeutic option for AD. 

One study of empagliflozin (EMP), an SGLT2 inhibitor, demonstrated decreased neuronal loss and improved cognition with an overall reduction in the soluble and insoluble Aβ levels in the cortex and hippocampus of EMP-treated mice of the APP/PS1xd/db model [159]. Another study of EMP-treated db/db mice resulted in a reduction in cognitive decline in EMP-treated mice [160]. Abnormal cholinergics have been linked to AD pathology and progression [161,162]. Acetylcholinesterase (AchE) is an important target for the treatment of AD. To that effect, canagliflozin, another SGLT2 inhibitor, improved memory dysfunction in rats with scopolamine-induced memory impairment [163], likely through a reduction in the acetylcholinesterase (AchE) activity, increased monoamines and acetylcholine M1 receptor (M1 mAchR). Another study was conducted to examine canagliflozin’s inhibitory effect on AchE, going so far as to dub it a dual inhibitor of AchE and SGLT2 [164].

One population-wide cohort study demonstrated a lower risk of dementia in SGLT2 inhibitor-treated elderly T2DM participants in comparison to the DPP4 inhibitor group [165]. Dapagliflozin exhibited the lowest risk, followed by empagliflozin; however, canagliflozin showed no association [165]. A nested case–control study showed a 42% decreased risk of dementia in T2DM patients [166]. Furthermore, a longitudinal study found a link between long-term (3+ years) SGLT2 inhibitor use on geriatric diabetics and improved cognitive function, as was measured by the Repeatable Battery for the Assessment of Neuropsychological Status (RBANS) [167]. These results indicate further investigations into the possible role SGLT2 inhibitors can play in slowing cognitive decline and, more to the point, the role they can play in AD management. 

## 4. Conclusions

Antidiabetics have a proven efficiency not only in the symptomatic management of AD, but also in attenuating disease progression. We have outlined multiple mechanisms through which these drugs function to therapize and improve the clinical course of AD both in vitro and in vivo; however, the central fact across each mechanism remains the same and that is the antidiabetic’s ability to target and ameliorate the key pathologies of AD: amyloid beta and tau hyperphosphorylation. There is also an improvement in cholinergic pathways and reduction in neuronal death, increased synaptic plasticity, and decreased neuroinflammation. Despite these encouraging results, there is still a long way to go in the development of anti-AD drugs, and further studies are necessary to confirm these drugs’ therapeutic potential. Additionally, new delivery approaches for medications like insulin are being planned which might offset their systemic effects [168]. Given the mechanistic overlap between T2DM and AD, it is logical to pursue anti-diabetic medications to treat AD and their effects are likely to be synergistic with other classes of medications such as cholinesterase inhibitors and anti-amyloid monoclonal antibodies. The data to date are encouraging and will further stimulate research.

**Table 1 biomedicines-12-00099-t001:** Summary of the clinical outcomes for diabetic drugs for AD.

Classes of Diabetic Drugs Being Repurposed for AD	Specific Drugs in the Class Being Explored for AD	Summary of Clinical Studies
metformin		A cohort study from Singapore in 365 participants showed that long-term metformin use was associated with a 50% risk reduction for cognitive decline A Finnish case–control study indicated that long-term use of metformin was associated with lower incidence of AD A pilot clinical trial in MCI showed improvements on the SRT
Insulin		Administering 20 IU intranasal insulin twice a day for 21 days improves attention, story recall, and function in those with MCI or AD Craft et al. ran a longer pilot clinical trial to investigate the effect of intranasal insulin on cognition, function, cerebral glucose metabolism, and CSF biomarkers in 104 adults with either AMCI (*n* = 64) or mild to moderate AD (*n* = 40) over the span of 4 months. The outcomes demonstrated an improvement in delayed memory in the 20 IU group (*p* < 0.05), as well as preserved caregiver-rated functional status at both insulin doses (*p* < 0.01) and general cognition assessed by the ADAS-cog score in younger patients and functional abilities assessed by the ADCS-ADL scale for adults with AD (*p* < 0.05) Intranasal insulin was used in a randomized clinical trial of 289 adults with MCI or AD over a span of 12 months; it showed no cognitive or functional benefits compared to the placebo group but there were lingering concerns about the delivery system
Incretins/GLP1 Receptor Agonists	Liraglutide, semaglutide, exenatide	ELAD, Evaluation of Liraglutide in the treatment of Alzheimer’s Disease, was a phase IIb double blinded, randomized, placebo-controlled trial conducted in multiple centres in the UK, where 204 adults with mild to moderate AD received subcutaneous injections of either LRGT or placebo once daily for 12 months The results demonstrated no difference between treatment and control in cerebral glucose metabolic rate, the primary endpoint [104]; however, there was improved cognitive function in LRGT-treated participants measured by ADAS-EXEC (ADAS-Cog with Executive domains of the Neuropsychological Test Battery). Currently, two large phase III clinical trials, evoke and evoke+ are underway. Each study has 1840 amyloid-positive participants with MCI or mild AD dementia who will be randomized to receive either daily oral semaglutide (14 mg, escalated via 3- and 7- mg over 8 weeks) or daily oral placebo over a period of 156 weeks The REWIND trial was a randomized, double-blind placebo-controlled trial conducted in 24 countries which examined the effect of once weekly subcutaneous injection of either DGT or placebo in participants aged 50 or more and diagnosed with T2DM on cardiovascular risks of T2DM, such as non-fatal MI, non-fatal stroke, or death from cardiovascular causes. After adjustment, the hazard of substantial cognitive impairment was reduced by 14% in the DGT-treated arm in comparison to placebo arm (HR 0.86 95% CI: 0.79–0.95 *p* = 0.0018), indicating that DGT may be a potential drug used to curb MCI in T2DM
DPP4 Inhibitors	Linagliptin, sitagliptin, saxagliptin, vildagliptin	A 2-year study of DPP4 inhibitors preserved cognition in diabetics with MCI A randomized trial of sitagliptin in 253 with T2DM showed MMSEs improved in the sitagliptin group
p-par gamma agonists	Rosiglitazone, piaglitazone	Piaglitazone associated with improved glucose metabolism and blood flow Rosiglitazone pilot associated with improved attention and delayed recall and ADAS. Efficacy signal selectively in ApoE 4 noncarriers Large phase III trial of rosiglitazone did not show a sustained efficacy signal
SGLT2 Inhibitors	Empagliflozin, cangliflozin, dapagliflozin	SGLT2 inhibitors lowered dementia risk by 42% Long-term improvement on the RBANS

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
