# Peer review of "Pharmacological Approaches Using Diabetic Drugs Repurposed for Alzheimer’s Disease"

_biomedicines, 2024, doi:10.3390/biomedicines12010099_

Round 1

Reviewer 1 Report

Comments and Suggestions for Authors

           The authors present an extensive narrative review regarding the positive impact of the most used antidiabetic drugs in AD. The paper is comprehensive and easy to read. Please find below a few comments for your consideration:

·       The aim of the study should be presented in the Abstract

·       “Introduction”: authors should mention that the antidiabetics presented in the manuscript are the most used in therapy according to ADA and EASD recommendations

https://diabetesjournals.org/care/article/45/11/2753/147671/Management-of-Hyperglycemia-in-Type-2-Diabetes

·       “Literature search” section: Please revise if terms are correct (e.g., “type 3 diabetes”, “ncretins Alzheimer trials”, “insulin resistance in the brain”)

·       A list of abbreviations would be useful.

·       I recommend adding a table with the results of clinical trials

Author Response

We appreciate the reviewers’ comments and have revised the manuscript accordingly. We have addressed each point, as detailed below. We believe that these changes help clarify the content of our paper and make it worthy of publication.

REVIEWER #1

 The authors present an extensive narrative review regarding the positive impact of the most used antidiabetic drugs in AD. The paper is comprehensive and easy to read. Please find below a few comments for your consideration:

  • The aim of the study should be presented in the Abstract  ADDED
  • “Introduction”: authors should mention that the antidiabetics presented in the manuscript are the most used in therapy according to ADA and EASD recommendations ADDED

https://diabetesjournals.org/care/article/45/11/2753/147671/Management-of-Hyperglycemia-in-Type-2-Diabetes

  • “Literature search” section: Please revise if terms are correct (e.g., “type 3 diabetes”, “incretins Alzheimer trials”, “insulin resistance in the brain”)

This section was reviewed per reviewer request

  • A list of abbreviations would be useful.

All abbreviations defined before being used

  • I recommend adding a table with the results of clinical trials

Table Added

Reviewer 2 Report

Comments and Suggestions for Authors

1- The manuscript suggests a link between insulin resistance in the periphery and the development of type 2 diabetes mellitus (T2DM), while also positing that insulin resistance in the brain plays a key role in Alzheimer's disease (AD) pathogenesis. Could the authors provide more detailed insights into the mechanistic aspects of insulin resistance in both peripheral tissues and the brain and how these processes contribute to the respective diseases?

2-The term "type 3 diabetes" is used to describe Alzheimer's disease. Can the authors elaborate on the molecular and physiological basis for this terminology, and discuss how this characterization informs our understanding of the relationship between T2DM and AD?

3-Could the authors provide a comparative analysis of the mechanisms of action of different classes of drugs mentioned in the text and how they may impact AD pathogenesis differently?

4- To improve the introduction and some parts of the discussion, the authors can use the following reference: https://doi.org/10.1016/j.ijbiomac.2022.01.134

5- How do the potential benefits of antidiabetic therapies in AD align with existing treatment approaches for AD, and are there any synergies or conflicts with standard AD medications?

Comments on the Quality of English Language

Minor editing of English language required.

Author Response

1- The manuscript suggests a link between insulin resistance in the periphery and the development of type 2 diabetes mellitus (T2DM), while also positing that insulin resistance in the brain plays a key role in Alzheimer's disease (AD) pathogenesis. Could the authors provide more detailed insights into the mechanistic aspects of insulin resistance in both peripheral tissues and the brain and how these processes contribute to the respective diseases?      

A clarifying sentence added to the introduction

2-The term "type 3 diabetes" is used to describe Alzheimer's disease. Can the authors elaborate on the molecular and physiological basis for this terminology, and discuss how this characterization informs our understanding of the relationship between T2DM and AD?     

A clarifying sentence added to the introduction

3-Could the authors provide a comparative analysis of the mechanisms of action of different classes of drugs mentioned in the text and how they may impact AD pathogenesis differently?   

A table was added per reviewer 1 suggestions.

4- To improve the introduction and some parts of the discussion, the authors can use the following reference: https://doi.org/10.1016/j.ijbiomac.2022.01.134       

Added

5- How do the potential benefits of antidiabetic therapies in AD align with existing treatment approaches for AD, and are there any synergies or conflicts with standard AD medications?  

A sentence was added to the discussion

Reviewer 3 Report

Comments and Suggestions for Authors

The document is organized and approaches a highly interesting topic. During my reading, I identified a few issues that could be enhanced to improve the quality of the manuscript, which follows:

Keywords should be improved by using synonyms or related terms, avoiding repeating words that were used in the abstract;

Introduction – It is interesting and well-designed; the background is adequate to support the hypothesis of the study; the objectives are sound;

Methods – Inclusion and exclusion criteria must be informed; The total amount of studies yielded from the search could be included, as well as the number of studies used to construct the review;

The document is excellently crafted and straightforward to navigate, with all the research thoroughly examined to accomplish the review's objectives. Nonetheless, it would be even more beneficial if the authors incorporated more interactive elements, such as tables and images. One suggestion is to create a table that condenses the primary findings, drugs experimented, animal models, and other pertinent data that would supplement the conversation. Additionally, the authors could seek authorization to use the illustrations from the referenced studies to enhance comprehension. Finally, it is strongly advised that the authors include an image that summarizes the principal findings of the study.

Author Response

The document is organized and approaches a highly interesting topic. During my reading, I identified a few issues that could be enhanced to improve the quality of the manuscript, which follows:

Keywords should be improved by using synonyms or related terms, avoiding repeating words that were used in the abstract;                                                   

 Keywords were expanded and added to the title pages

Introduction – It is interesting and well-designed; the background is adequate to support the hypothesis of the study; the objectives are sound;                            Thank you

Methods – Inclusion and exclusion criteria must be informed; The total amount of studies yielded from the search could be included, as well as the number of studies used to construct the review;                                                

We added clarifying language per reviewer suggestion

 The document is excellently crafted and straightforward to navigate, with all the research thoroughly examined to accomplish the review's objectives. Nonetheless, it would be even more beneficial if the authors incorporated more interactive elements, such as tables and images. One suggestion is to create a table that condenses the primary findings, drugs experimented, animal models, and other pertinent data that would supplement the conversation. Additionally, the authors could seek authorization to use the illustrations from the referenced studies to enhance comprehension. Finally, it is strongly advised that the authors include an image that summarizes the principal findings of the study.            

A table was added per reviewer # 1 recommendations